# Influence of Sex, Family Structure, and Access to Technology on the Motor Development of Children Aged 24 to 48 Months

**DOI:** 10.3390/healthcare13243191

**Published:** 2025-12-05

**Authors:** Miguel Rebelo, Rafael Adrião, Marco Batista, Samuel Honório, Helena Mesquita, Catarina Marques, João Serrano

**Affiliations:** 1Department of Sports and Well-Being, Polytechnic Institute of Castelo Branco, 6000-266 Castelo Branco, Portugal; rafael-adriao@hotmail.com (R.A.); marco.batista@ipcb.pt (M.B.); samuelhonorio@ipcb.pt (S.H.); hmesquita@ipcb.pt (H.M.); catarina.marques3@ipcb.pt (C.M.); j.serrano@ipcb.pt (J.S.); 2SPRINT Sport Physical Activity and Health Research & Innovation Center, 6000-084 Castelo Branco, Portugal; 3Centro Interdisciplinar de Ciências Sociais, Universidade Nova, 1070-312 Lisboa, Portugal

**Keywords:** motor development, PDMS-2, sex, sibling presence, duration of access to technology

## Abstract

**Background/Objectives**: This study aimed to analyze the influence of sex, the presence of siblings, and the duration of exposure to technology on the development of gross and fine motor skills in children aged between 24 and 48 months, using the PDMS-2 battery as the assessment instrument. **Methods**: The sample comprised 193 children, distributed across three age groups: 24 months (N = 22), 36 months (N = 78), and 48 months (N = 93). The assessed skills included Postural Control, Locomotion, Object Manipulation, Fine Grasping, and Visual–Motor Integration, grouped into the domains of Gross Motor and Fine Motor development. Statistical analysis was conducted using the non-parametric Mann–Whitney and Kruskal–Wallis tests, complemented by the epsilon squared (ε^2^) effect size measure. **Results**: The results revealed statistically significant differences between sexes, with girls demonstrating superior performance in Fine Motor and Visual–Motor Integration tasks, particularly at 36 and 48 months (*p* < 0.05; ε^2^ up to 0.22). The presence of siblings showed a positive impact only at 36 months, while the duration of exposure to technology did not present a significant association with motor performance at any age group. **Conclusions**: The Total Motor Quotient (TMQ) varied according to the variables analyzed, reinforcing the notion that motor development is multifactorial and sensitive to familial and social contexts. These findings highlight the importance of considering both environmental and biological factors when designing motor intervention strategies in early childhood.

## 1. Introduction

Motor development in early childhood is widely recognized as a dynamic and multifactorial process, influenced by biological, environmental and social factors [1,2]. Between the ages of 24 and 48 months, children go through a critical phase of acquiring fundamental motor skills, which form the basis for physical, cognitive and social performance in later stages [3]. These skills fall into two broad categories: gross motor skills, which involve broad movements and large muscle groups, and fine motor skills, which require precision, hand–eye coordination and control of small muscles [4]. Scientific literature has shown that motor development does not occur homogeneously among children, but is influenced by variables such as sex, family structure and educational context. These variables can accelerate or hinder the acquisition of motor skills, and it is essential to understand them in order to promote effective and equitable interventions [3,5].

Sex is one of the most studied variables in motor development. Several longitudinal studies indicate that boys tend to perform better in gross motor skills, such as running and jumping, while girls excel in fine motor tasks, such as drawing or manipulating small objects [6,7,8]. Adaptations of the Portuguese version of the Peabody Developmental Motor Scales—Second Edition (PDMS-2) found that girls achieved higher results in fine motor and visuomotor integration subtests, while boys excelled in global motor tasks [9]. Differences between sexes have also been observed in children aged 36 to 71 months, with girls demonstrating greater precision in manual control and visuomotor coordination tasks [9]. However, other studies involving younger children (0–18 months) found no significant sex-based differences in total scores or posture assessments, suggesting that distinctions may emerge later and be shaped by environmental context and practice opportunities [10]. A systematic review of studies involving children aged 3 to 10 years revealed that 61% reported no significant sex differences, while the remaining studies showed alternating superiority depending on the motor skill assessed [11]. These findings highlight that the literature is mixed: while female advantages are often reported in fine motor and visuomotor domains, gross motor differences are less consistent and appear strongly influenced by practice opportunities and sociocultural norms [12,13,14]. Thus, gender differences in motor development should be interpreted as the result of an interaction between biological maturation, neuromotor development, and sociocultural play styles.

Sibling presence is another variable that can significantly impact motor development. Children with siblings are often exposed to shared play, competition, and imitation, which may accelerate motor skill acquisition [15]. Older siblings can serve as role models, promoting challenges and practice opportunities not available to children without siblings [16]. On the other hand, the absence of siblings can allow for greater individualized attention from caregivers, which may also favor motor development, especially in tasks requiring direct supervision [17]. Research involving children up to 48 months of age found that those with siblings performed better on average in global motor tasks, supporting the hypothesis that sibling interaction fosters motor practice and physical challenges [18]. Sibling relationships are often emotionally intense and characterized by frequent physical play, symbolic games, and spontaneous motor challenges, which contribute to the development of coordination, balance, and postural control [19,20]. These interactions, especially when older siblings assume leadership roles, promote movement diversity and repetition, refining fundamental motor skills. Nonetheless, the absence of siblings does not necessarily imply motor delays, as stimulating family environments with attentive caregivers can also support high levels of motor development [21]. The impact of sibling presence may vary depending on sex and age gap, with girls often benefiting more in fine motor tasks and boys in gross motor activities [16].

Technology use in early childhood has also been studied as a variable with potential impact on motor development, given the increasing exposure of children to digital devices. Although technologies can offer visual stimuli and cognitive challenges, their effect on motor skills depends on duration, type of game, and balance with physical activity. Passive technologies, which require little physical interaction, have been linked to deficits in gross motor skills, whereas interactive technologies can stimulate fine motor development [22,23,24]. Controlled exposure to action-based digital games has been shown to improve reaction time and manual precision, while excessive screen time, particularly when replacing outdoor play, poses risks [22,25,26]. Preschool children who spend more time on digital devices have shown lower performance in strength and manipulation tests, reinforcing concerns about sedentary behavior replacing spontaneous motor activity [27]. These findings underscore the importance of considering not only the duration but also the type and context of technology use.

This study is relevant as it addresses motor development during early childhood—a critical phase for physical and cognitive growth. By examining sex, sibling presence, and technology exposure, it offers a multifactorial analysis that reflects current concerns about digital influence and family dynamics. The theoretical foundation, grounded in established literature on psychomotor development, supports the scientific validity of the research. In addition, the study has significant practical applicability, contributing to more effective pedagogical interventions and family guidance. The inclusion of technology exposure as a variable is particularly timely, given the increase in digital use among young children, while the exploration of sibling presence broadens understanding of social stimuli that affect motor development.

The aim of the study is to analyze whether differences in motor skills arise when considering sex, the presence of siblings, and exposure to technology in children aged 24 to 48 months. Based on the mixed evidence in the literature, we expected that sex differences might be limited or inconsistent in this age group, although previous studies have reported female advantages in fine motor and visuomotor domains. We also expected that children who live with siblings would show superior motor development due to increased social stimuli and opportunities for physical interaction in the family environment. Finally, we hypothesized that children who spend less time using technology daily would demonstrate better motor skills, both gross and fine, compared to those with greater digital exposure.

## 2. Materials and Methods

This study is characterized as quantitative, descriptive and cross-sectional, using non-probabilistic convenience sampling. The study took place in kindergartens and nurseries in the municipalities of Rio Maior and Santarém.

The sample consisted of a total of 193 children aged between 24 and 48 months (40.41 ± 8.16). The sex variable (1.51 ± 0.50) consisted of 95 male children and 98 female children. When characterizing the variable considering the children’s age in months, males (40.17 ± 7.97) and females (40.65 ± 8.37). Regarding the presence of siblings (1.31 ± 0.46), 133 children had siblings, while 60 children had no siblings at all. Considering the number of siblings (1.04 ± 1.03), 86 children have only one sibling (44.6%), while 47 children have between 2 and 6 siblings (24.4%). Considering the time per day that the child has access to technology (1.46 ± 0.49), 105 children spend less than 60 min per day in contact with technology (54.4%), where 88 children spend more than 60 min per day interacting with technology (45.6%).

To select the sample, the following inclusion criteria were defined: children aged between 24 and 48 months. Exclusion criteria: children diagnosed with learning difficulties, developmental disorders and also with some kind of diagnosed disability, respectively.

The decision to divide the sample into three age groups (24, 36, and 48 months) was based on the normative structure of the PDMS-2, which provides age-specific cut-off points and reference values. This categorization allowed us to align the analysis with the technical framework of the instrument, facilitating clinical interpretation and comparability with previous studies. Although motor development is inherently continuous, the use of normative age groups ensured methodological consistency and enhanced the robustness of the comparisons performed.

### 2.1. Instrument

The instrument used to collect information on the motor profile and motor skills of the children in the study was the PDMS-2 [28]. The PDMS-2 scales are one of the most widely used instruments in motor assessment to check the motor profile of children from birth to 71 months, and these scales were translated for the Portuguese population by Saraiva et al. (2011) [29] and for the ages under study (χ^2^ = 55.614; gl = 4; *p* = 0.06; χ^2^/gl = 13.904; SRMR = 0.065; CFI = 0.99; TLI = 0.99; α = 0.85 and ICC = 0.98).

Studies state that the composite structure of the PDMS-2 includes five subtests divided into two motor components: global motor skills and fine motor skills [18,29].

Their results are expressed in three domains of motor behaviour: the fine motor quotient (FQM), the global motor quotient (GMQ) and the total motor quotient (TMQ), the latter resulting from the first two and indicating the motor profile. The FQM is found by adding up two sets of subtests, fine grip and visuomotor integration, while the GMQ uses three subtests, postural control, locomotion and object manipulation. Each of these subtests is made up of motor tasks adjusted to age and placed in an increasing sequence of difficulty [29].

The current version has other advantages which specifically allow the assessment of the child’s motor competence in relation to their peers, the identification of motor deficits and imbalances between the fine and global motor domain, the establishment of individual goals and objectives in clinical intervention and the monitoring of the child’s individual development, with the ability to classify the child’s level according to their age, on a scale ranging from “very poor” to “very good” [18].

### 2.2. Data Collection Procedure

After institutional approval, an informed consent form was given to the parents/guardians, who were also asked to fill in the child’s characterization form. All ethical principles and international standards, namely the Declaration of Helsinki and the Convention on Human Rights and Biomedicine, were followed, respected and preserved, and the study was approved by the Ethics Committee No. 184/CE-IPCB/2024 [30].

To obtain information about the participants, an anamnesis form was created, in which information necessary for the study was collected [31].

The child starts the test on a specific item, according to their age, and continues in the sequence until they fail three consecutive executions. Each item is classified according to an evaluation scale with three values: 0 = no performance, 1 = minimum proficiency, 2 = optimum proficiency. The value of the sum of all the items in each of the subtests is located in the reference table for age, resulting in a standardized value and a percentage value that can be compared between ages [18].

The items are added together in each of the tests and their value is located in the reference table for age. Then, the sum of the standardized values of the grouped tests makes it possible to obtain the TMQ, GMQ and FMQ by consulting an appropriate table. Subsequently, the value of the sum of the items in each of the sub-scales is located in a reference table for age, where a standardized value (from 1 to 20) is obtained, which has been converted into a qualitative classification with seven categories (from 1—“Very Poor” to 7—“Very Good”) [29,32].

For Folio and Fewell (2000) [28], the administration of the PDMS-2 must be based on the assumption of the original authors, in which examiners must understand the general procedures for administering the test, its scoring and interpretation of results. The battery will be administered individually and for 45 to 60 min, in a large room or space with stairs. The assessment location will be prepared in advance to provide an environment with as few stimuli and distractions as possible. The time the test is administered will respect school routines, meal times and all other times considered important and mandatory by the schools. The assessments, when interrupted, will be completed within five days, as established by the authors of the instrument [18,29].

In order to correctly administer the instrument, we followed the guidelines of Saraiva et al. (2011) [29]:Instructions repeated to the child 3 times in order to provide the opportunity to achieve the maximum score on each item;The child starts the test at a point on the scale established by their age and continues in sequence until they fail to complete three consecutive items;The scores for each item are between 0 and 2.

### 2.3. Statistical Procedures

IBM SPSS Statistics software, version 20, was used to analyze the data statistically. Initially, descriptive statistics were used to calculate measures of central tendency and dispersion, namely: mean, median, minimum, maximum and standard deviation. Before applying inferential tests, the normality of the data distribution was assessed, an essential criterion for choosing the most appropriate statistical tests.

Lercas (2018) [33] considers that normality tests are used to check whether the observations in the sample fit their distribution, and there are two types of tests to check normality: Kolmogorov–Smirnov and Shapiro–Wilk. The Kolmogorov–Smirnov test was used, with all the variables chosen from the sample having a non-normal distribution (*p* < 0.05).

The choice of subsequent statistical tests will therefore be determined on the basis of the results of the normality analysis, guaranteeing the methodological suitability and validity of the results obtained.

In order to draw conclusions from this study, we first carried out a descriptive analysis of the motor variables by age group. Due to the ordinal nature of the data and the verification of deviations in tests of normality and homogeneity of variance, we used the Mann–Whitney U-test for comparisons between two independent groups and the Kruskal–Wallis test for comparisons involving three or more groups, as these non-parametric methods dispense with strict assumptions of normal distribution and are robust in samples of varying size. In addition, the effect size ε^2^ (epsilon square) was calculated to estimate the practical magnitude of the differences observed, since ε^2^ provides a less biased estimate in non-parametric contexts and avoids the typical overestimation of η^2^ (eta square) in samples [34,35]. The interpretation of ε^2^ values followed the guidelines of Tomczak and Tomczak (2014) [35]: 0.00–0.20 (small), 0.21–0.60 (medium) and >0.60 (large), complemented by correlation scales adapted from Rea and Parker (2005) [36], for a more refined analysis of the strength of associations. Given the exploratory nature of the study, no formal correction for multiple testing was applied.

## 3. Results

To answer the question and hypotheses, which seeks to verify whether there are differences in motor skills considering the variables sex, presence of siblings and time of access to technologies, statistical tests were carried out on PDMS-2 data, distributed over three age groups: 24, 36 and 48 months. The specific hypotheses propose that there are no differences between the sexes in motor development, that children with siblings have better motor performance, and that less time spent using technology on a daily basis is associated with better motor skills. Table 1, Table 2 and Table 3 allow us to assess the validity of these hypotheses and understand the impact of each variable on children’s motor development.

Table 1 shows that, contrary to the initial hypothesis, there are statistically significant differences between the sexes in various motor skills, especially at the ages of 24 and 48 months. At 24 months, girls performed better in postural control (*p* = 0.030) and object manipulation (*p* = 0.050), with medium size effects (ε^2^ = 0.22 and 0.19), indicating a moderate influence of sex on these skills. At 36 months, visuomotor integration (*p* = 0.003) and fine motor skills (*p* = 0.003) stood out, again with an advantage for girls and small size effects (ε^2^ = 0.11), suggesting greater precision and manual coordination in this group.

At 48 months, the differences became even more evident: girls outperformed boys in postural control (*p* = 0.001), visuomotor integration (*p* = 0.019), global motor skills (*p* = 0.001) and fine motor skills (*p* = 0.049). These results indicate that, with advancing age, the differences between the sexes become more pronounced in specific motor domains, contradicting the hypothesis of no differences. The size effect varies between small and medium, but is consistent, reinforcing the relevance of sex as an influential variable in motor development.

Thus, the data in Table 1 allows us to reject the study’s null hypothesis in relation to sex, showing that girls tend to perform better in tasks that require postural control and visuomotor integration, while boys do not show significant superiority in any of the sub-scales assessed. These results are in line with some of the literature that points to sex differences in fine motor and visual skills.

Table 2 analyzes the impact of the presence of siblings on motor development. At 24 months, there were no significant differences between children with and without siblings in any of the subscales, with high *p*-values and no or very low effect sizes. This suggests that, at this early stage, the presence of siblings has no relevant influence on motor performance, possibly due to the limited structured motor interaction between siblings at this age.

At 36 months, there were significant differences in visuomotor integration (*p* = 0.001) and fine motor skills (*p* = 0.006), with higher averages in children with siblings and average size effects (ε^2^ = 0.13 and 0.09). These data indicate that living with siblings can enhance the development of skills that require coordination and manipulation, probably through shared play and imitation of motor behaviors. Object manipulation also showed a significant trend (*p* = 0.085), reinforcing this interpretation.

At 48 months, the differences were again non-significant, with *p*-values > 0.1 in all subscales. Fine and Global Motor Skills showed similar averages between the groups, suggesting that the impact of the presence of siblings may be more relevant in a specific window of development, namely at 36 months. These results partially validate the hypothesis that children with siblings have better motor performance, especially in fine and visuomotor skills.

Table 3 evaluates the influence of daily technology time (<60 min or >60 min) on motor skills. At 24 months, there were no statistically significant differences, although children with more than 60 min showed higher averages in locomotor skills and object manipulation. However, the small sample size (N = 5) limits the interpretation of this data, and the effect sizes are low.

At 36 months, the results remain non-significant, with high *p*-values in all subscales. Curiously, Fine Motor Skills shows slightly higher averages in children with more screen time, but this is not statistically significant. This may indicate that, in this age group, screen time does not directly compromise motor development, as long as it is balanced with other activities.

At 48 months, the data still showed no significant differences. The averages are similar between the groups, with slight advantages for children with longer access to technologies in Global and Fine Motor Skills. The size effects are low, suggesting that time of use is not an isolated determining factor. These results do not support the hypothesis that less time accessing technology is associated with better motor performance, at least in the age groups analyzed.

In this way, we can conclude that the results of the tables analyzed provide a clear answer to the hypotheses of this study. There are statistically significant differences between the sexes in various motor skills, with girls performing better in areas such as fine motor skills, visuomotor integration and postural skills, especially at 36 and 48 months. The results partially confirm the study’s hypotheses: sex and the presence of siblings influence motor development in early childhood, while time of access to technology does not prove to be a determining factor. These findings are supported by the skill averages, statistical significance values and size effects observed, allowing for a deeper understanding of the factors that shape the motor profile of children between 24 and 48 months of age.

## 4. Discussion

The analysis of the data reveals important nuances about motor development in early childhood, considering three crucial variables: sex, presence of siblings and time of access to technology. The results obtained allow for a critical reflection on the factors that influence motor skills in children between 24 and 48 months of age.

Initially, the sex variable proved to be a significant factor in several subscales of the PDMS-2, contradicting the study’s null hypothesis. Girls performed better in postural skills, fine motor skills and visuomotor integration, especially at 36 and 48 months. According to Zheng et al. (2022) [37], girls tend to perform better in fine motor and balance tasks, which is in line with the data in this study. The meta-analysis conducted by these authors confirms that sex differences are consistent in children between 36 and 72 months of age, with an advantage for females in fundamental motor skills.

Although our initial hypothesis assumed limited or inconsistent sex differences, based on contradictory evidence in the literature, the results revealed clear and consistent female advantages in fine motor and visuomotor domains. This outcome aligns with studies reporting earlier neuromotor maturation in girls and sociocultural play styles that favor precision and coordination tasks [6,7,8,9,37]. Thus, the findings reinforce the importance of considering sex as a relevant factor in early motor development. The consistency of the female advantage observed in our sample supports previous findings, while also clarifying that such differences may become more evident between 36 and 48 months, a period marked by rapid neuromotor maturation. This female advantage can be explained by earlier neuromotor maturation, which facilitates fine motor milestones, combined with sociocultural play styles that often emphasize precision tasks for girls. These factors interact to produce consistent differences in fine motor and visuomotor integration at 36 and 48 months.

The superiority of girls in tasks that require precision and coordination may be related to earlier neuromotor development observed in longitudinal studies. For example, Haywood and Getchell (2021) [2] point out that girls tend to reach fine motor milestones earlier than boys, which may explain the results observed at 24 and 36 months. In this study and on the specific tasks assessed, boys did not outperform girls. This finding should not be interpreted as a general lack of ability, but rather as evidence that, within the PDMS-2 subscales applied, no male advantage was observed. Several factors may explain why boys did not show the gross motor advantages often reported in other studies. First, biological maturation trajectories differ, with girls often reaching fine motor milestones earlier, which may overshadow male performance in certain domains at these ages. Second, the types of play encouraged in early childhood can vary by sex, with girls often engaging in activities that promote precision and coordination, while boys’ gross motor play may not be fully captured by the PDMS-2 tasks. Finally, test item sensitivity may play a role, as some gross motor skills where boys typically excel (e.g., running speed, jumping distance) may not be emphasized in the subscales analyzed. These considerations highlight that the absence of male advantage in our results reflects task-specific and developmental factors rather than a general lack of motor competence. This interpretation is consistent with Barnett et al. (2009) [3], who emphasize that motor development is heterogeneous and shaped by sociocultural opportunities, and with Westendorp et al. (2011) [38] and Piek et al. (2008) [39], who found female advantages in manual coordination and visuomotor tasks. Lopes et al. (2011) [40] further highlight that sociocultural play styles modulate these differences, reinforcing the multifactorial nature of motor development.

The presence of siblings only had a significant impact at 36 months, particularly on visuomotor and fine motor skills. This finding suggests that living with siblings promotes opportunities for more complex motor interactions, such as cooperative play and imitation of movements. According to Meltzoff (2007) [41,42], imitation is a powerful learning mechanism in childhood, and children with older siblings tend to replicate observed motor behaviors, which may explain the superior performance observed at this age. The data from this study also shows that children with siblings achieved higher mean scores in manipulation and coordination tasks, corroborating Haywood and Getchell (2009) [43], who emphasize the role of family interactions in motor development, especially during sensitive developmental windows. This sensitive window may also be explained by developmental milestones that typically peak around 36 months. At this age, children begin to engage in more complex cooperative play, which requires negotiation, turn-taking, and coordinated motor actions, often facilitated by sibling interaction [43,44]. Language development also accelerates, enabling richer communication and joint problem-solving during play, while imitation and role-taking become more sophisticated, allowing younger children to replicate and adapt the motor behaviors of older siblings with greater accuracy [41]. These combined advances in social, linguistic, and motor domains may amplify the influence of sibling interactions specifically at 36 months.

The fact that the effect was strongest at 36 months may reflect a critical motor period, during which children are particularly receptive to role modeling and imitation. In contrast, at 24 months children may still be too immature to benefit fully from sibling interaction, while at 48 months motor skills may already be more consolidated, reducing the relative impact of sibling presence. This interpretation is consistent with Cools et al. (2009) [44], who argue that the influence of siblings depends on the developmental stage and the quality of interactions, which may explain the variability of the results across age groups.

With regard to time spent accessing technology, the data does not support the hypothesis that less time is associated with better motor performance. In all age groups, there were no statistically significant differences between the groups with less or more than 60 min of daily use. This result contradicts some of the literature that associates screen time with delays in motor development. For example, Tremblay et al. (2011) [45] warn of the risks of sedentary lifestyles induced by digital activities, especially at an early age. However, the data from this study suggests that the amount of time spent using technology does not, in itself, directly affect motor skills. One possible explanation for this lack of impact may lie in the nature of the technologies used. Studies such as Staiano and Calvert’s (2011) [46] show that the use of active technologies, such as those involving body movement, can even promote motor development, especially in coordination and balance tasks. Carson et al. (2016) [47] suggest that the negative impact of screen time is mitigated when children regularly participate in structured motor activities, such as outdoor play or physical education programs.

It is important to emphasize that the binary categorization of technology exposure (<60 vs. >60 min) represents a methodological oversimplification. This measure does not distinguish between passive screen time (e.g., watching videos) and interactive technologies (e.g., touch-screen games requiring fine motor input), which may exert distinct influences on motor development. This lack of nuance likely contributed to the non-significant findings, as interactive technologies could support fine motor skills, while passive exposure may reduce opportunities for gross motor practice. The PDMS-2 subscales may also be less sensitive to capturing subtle benefits of interactive use. Future studies should therefore adopt more refined measures of technology use, considering modality, context, and parental mediation, to provide a more accurate understanding of its impact on motor development.

It should also be noted that in the 24-month group, the subgroup with >60 min of technology exposure comprised only five children (N = 5). This very small sample size prevents meaningful statistical comparison, and the trends observed in this subgroup cannot be considered reliable. This limitation may partly explain the absence of significant associations in this age group.

The interaction between the variables sex and the presence of siblings may be relevant. According to Stodden et al. (2008) [48], motor development is influenced by multiple interacting factors, including biology and family environment. Figueroa and An (2017) [49] highlight that the presence of siblings and opportunities for motor interaction significantly influences motor competence and physical activity levels in preschool children. The absence of an isolated impact of the time spent using technology reinforces the idea that motor development is multifactorial, depending not only on individual factors, but also on the family and social dynamics that shape the child’s motor experiences. Research such as that by Leblanc et al. (2012) [50] indicates that screen time should be analyzed in conjunction with other factors, such as physical activity and family stimuli. According to Payne and Isaacs (2017) [4], knowledge about the factors that influence motor development allows for more effective interventions in early childhood. International literature recommends promoting environments rich in motor stimuli, as advocated by Logan et al. (2012) [51], who highlight the importance of diversified motor experiences for children’s overall development.

According to Figueroa and An (2017) [49], the domestic context has a significant influence on motor competence in childhood, especially when there are frequent social and motor stimuli, such as play between siblings. In addition, Pereira et al. (2023) [11], in a systematic review, point out that although there are sex differences in some motor skills, these tend to be modulated by environmental and cultural factors, such as equal access to organized motor experiences. The discussion of the results must therefore consider the role of the socio-cultural environment in the expression of motor skills. Although the study was carried out in Portugal, the patterns observed are compatible with international evidence pointing to the influence of context in modulating sex differences, as demonstrated by Utesch et al., (2019) [52] in European populations.

The results contribute to understanding the variables that influence motor development in early childhood. The data confirms that sex and the presence of siblings influence motor development in early childhood, while in this sample a simple binary measure of daily technology duration (<60 vs. >60 min) was not associated with motor performance. These findings suggest that the type and context of technology use are likely more critical factors than duration alone.

These conclusions must, however, be interpreted with caution given the methodological constraints. The cross-sectional design prevents causal inferences, meaning that the associations observed cannot be interpreted as developmental trajectories or cause–effect relationships. In addition, the use of convenience sampling, although common in studies with young children, limits the representativeness of the sample and therefore the generalizability of the results to wider populations. Children recruited from nurseries may differ in socioeconomic status, cultural background, and family stimulation compared to other populations, which restricts external validity. These factors reinforce that the conclusions drawn here should be considered preliminary and contextual, requiring confirmation in longitudinal studies with probabilistic sampling.

Another statistical limitation concerns the absence of multiple comparison corrections. Given the number of comparisons performed, this may increase the likelihood of Type I error, and the results should therefore be interpreted with caution. While the reporting of epsilon-squared effect sizes strengthens the analysis, it is important to note that small values indicate limited practical significance, even when statistically significant.

A further limitation concerns the measurement of technology exposure in binary categories (<60 min or >60 min per day), which oversimplifies a multifaceted practice. This approach does not distinguish between passive screen time (e.g., watching videos) and interactive technologies (e.g., touch-screen games requiring fine motor input), nor does it account for the diversity of game genres, levels of interactivity, platforms, or parental mediation. These qualitative aspects may exert distinct influences on motor skills and likely explain the non-significant findings observed in this study.

Another methodological consideration relates to the categorization of age into three groups (24, 36, and 48 months). While motor development is inherently continuous, this approach was adopted to align with the normative structure of the PDMS-2, which provides age-specific cut-off points and reference values. This ensured methodological consistency and facilitated comparability with previous studies, although future research may benefit from modeling age as a continuous variable to capture developmental trajectories more precisely.

For future research, we recommend the use of probabilistic or stratified sampling based on sociodemographic variables to increase external validity. A more detailed assessment of technology use should be implemented, ideally through validated questionnaires that record the type of technology, the context of use, and the presence of adult mediation. Longitudinal designs would allow mapping of motor trajectories and identification of predictors of specific outcomes, thereby distinguishing between cause and effect. In addition, exploring moderating variables such as sibling age differences, time spent in early childhood education, or parental physical activity levels could refine the understanding of the mechanisms shaping motor development.

By adopting these approaches, future studies may provide more robust evidence to guide interventions and policies that promote balanced motor development in increasingly diverse family contexts.

## 5. Conclusions

The results for the sex variables show that girls exhibit consistently superior performance in postural skills, visuomotor integration and fine motor skills, especially at 36 and 48 months, contradicting the initial hypothesis that there were no differences between boys and girls. Girls achieved significantly higher averages in the PDMS-2 subscales, with moderate effect sizes, indicating a more accelerated neuromotor development pattern in these specific skills.

In contrast, boys did not show a statistically significant advantage in any of the subscales evaluated, which reinforces the idea that fine motor and visual-manual development tends to favor females in early childhood. These results suggest the need to differentiate motor stimulation strategies according to sex, valuing activities that promote the refinement of manual and postural skills in boys.

As for the presence of siblings, there was a positive impact only at the age of 36 months, with significant values in fine motor skills and visuomotor integration for children living with siblings. This positive impact decreased at 48 months, indicating a sensitivity in which sibling interaction enhances motor learning opportunities, possibly through shared play and modeling of motor behaviours.

At 24 months, the presence of siblings did not influence motor performance, probably due to the still incipient nature of motor interactions between infants. By the end of the study period, the similarity of means between groups with and without siblings suggests that other socialization contexts, such as early childhood education and extracurricular activities, play a preponderant role.

Daily time spent on technology, divided into less than 60 min or more than 60 min, showed no statistically significant association with any of the motor domains assessed. Even in the averages where the group with more electronic exposure achieved a slight advantage, the low ε^2^ values and the lack of significance indicate that, in this sample, the sheer volume of screen time alone did not show meaningful associations with children’s motor profile. These findings should be interpreted cautiously, given the simplified measurement and the cross-sectional design, and considered exploratory until confirmed in future studies.

These results challenge the assumption that less time using technology necessarily guarantees better motor development, highlighting instead the importance of considering the quality and type of use, as well as the balance with physical activities and social interactions, before attributing generic negative impacts to the use of electronic devices.

## 6. Practical Implications

The findings of this study have substantial implications for healthcare systems, clinical practice, and public health policy. Motor development during early childhood is a cornerstone of lifelong health, influencing physical activity patterns, weight management, and cognitive performance. Integrating motor screening into routine pediatric care is essential for early detection of developmental delays. Health professionals should adopt standardized tools such as PDMS-2 to assess gross and fine motor skills during well-child visits, enabling timely interventions. Sex-specific trends observed in this study indicate the need to ensure that all children receive opportunities to strengthen postural control, visuomotor integration, and fine motor skills. Particular attention should be given to boys, who in this study tended to lag behind in postural and visuomotor domains, so that they can benefit from targeted activities that promote balanced motor development. Although screen time did not show a direct negative impact, healthcare providers should continue to advocate for balanced technology use, emphasizing active play and outdoor activities to mitigate sedentary behaviors. These recommendations can be operationalized through multidisciplinary programs involving pediatricians, physiotherapists, occupational therapists, and educators, ensuring a holistic approach to child development. At the policy level, incorporating motor development guidelines into national health strategies and early childhood education curricula can reduce disparities and promote equity. By fostering environments rich in motor stimuli and supporting families through education and counseling, healthcare systems can contribute to optimal developmental outcomes, reducing future risks of motor impairments, obesity, and associated chronic diseases. Ultimately, these actions align with global health priorities for early childhood and underscore the role of preventive care in shaping healthier generations.

## Figures and Tables

**Table 1 healthcare-13-03191-t001:** Differences between the Sex variable in PDMS-2 for each age group.

Age Group	PDMS-2	Sex	N	M ± SD	*p*	ε^2^	Effect Size
24 months	Posture Skills	Male	10	10.30 ± 1.89	0.030	0.22	1.013
		Female	12	12.17 ± 1.80			
	Locomotive Skills	Male	10	7.50 ± 2.22	0.497	0.02	0.229
		Female	12	8.00 ± 2.13			
	Object Handling Skills	Male	10	7.60 ± 1.57	0.050	0.19	0.843
		Female	12	8.92 ± 1.56			
	Fine Manipulation Skills	Male	10	10.70 ± 4.16	0.539	0.01	0.287
		Female	12	9.67 ± 2.90			
	Skills VM integration	Male	10	7.30 ± 2.35	0.140	0.11	0.654
		Female	12	9.00 ± 2.82			
	Global Motricity	Male	10	26.30 ± 5.31	0.107	0.13	0.799
		Female	12	30.00 ± 3.83			
	Fine motor skills	Male	10	17.20 ± 4.82	0.080	0.15	0.877
		Female	12	21.17 ± 4.21			
36 months	Posture Skills	Male	42	11.55 ± 2.83	0.271	0.16	0.239
		Female	36	12.17 ± 2.32			
	Locomotive Skills	Male	42	8.93 ± 1.04	0.377	0.00	0.215
		Female	36	9.17 ± 1.18			
	Object Handling Skills	Male	42	7.52 ± 1.48	0.632	0.00	0.050
		Female	36	7.44 ± 1.66			
	Fine Manipulation Skills	Male	42	10.48 ± 2.81	0.544	0.00	0.130
		Female	36	10.89 ± 3.42			
	Skills VM integration	Male	42	10.50 ± 2.08	**0.003**	**0.11**	**0.634**
		Female	36	**11.78 ± 1.95**			
	Global Motricity	Male	42	28.19 ± 3.95	0.940	0.00	0.042
		Female	36	28.36 ± 4.09			
	Fine motor skills	Male	42	21.26 ± 4.61	0.003	0.11	0.689
		Female	36	24.14 ± 3.70			
48 months	Posture Skills	Male	43	9.30 ± 2.50	**0.001**	**0.14**	**0.887**
		Female	50	**11.14 ± 1.53**			
	Locomotive Skills	Male	43	7.42 ± 1.31	0.075	0.03	0.390
		Female	50	7.98 ± 1.55			
	Object Handling Skills	Male	43	6.65 ± 0.75	0.221	0.01	0.284
		Female	50	6.90 ± 0.99			
	Fine Manipulation Skills	Male	43	10.60 ± 2.80	0.825	0.00	0.081
		Female	50	10.38 ± 2.61			
	Skills VM integration	Male	43	10.86 ± 2.00	**0.019**	**0.05**	**0.499**
		Female	50	**11.78 ± 1.67**			
	Global Motricity	Male	43	23.65 ± 3.82	**0.001**	**0.12**	**0.719**
		Female	50	**26.40 ± 3.39**			
	Fine motor skills	Male	43	20.60 ± 3.67	**0.049**	**0.04**	**0.496**
		Female	50	**22.50 ± 3.98**			

Significant values are in bold. N—sample number; M—Mean; SD—Standard Deviation; Min—Minimum; Max—Maximum; ε^2^: 0–0.2: small; 0.21–0.6: medium; >0.61: large.

**Table 2 healthcare-13-03191-t002:** Differences between the variable Presence of Sibling in the PDMS-2 for each age group.

Presence of Siblings	PDMS-2	Brother	N	M ± SD	*p*	ε^2^	Effect Size
24 months	Posture Skills	Yes	15	11.40 ± 1.95	0.782	0.00	0.120
		No	7	11.14 ± 2.34			
	Locomotive Skills	Yes	15	7.67 ± 1.95	0.731	0.00	0.142
		No	7	8.00 ± 2.64			
	Object Handling Skills	Yes	15	8.33 ± 1.58	0.891	0.00	0.022
		No	7	8.29 ± 1.97			
	Fine Manipulation Skills	Yes	15	10.47 ± 3.20	0.368	0.04	0.278
		No	7	9.43 ± 4.19			
	Skills VM integration	Yes	15	8.47 ± 2.74	0.535	0.01	0.276
		No	7	7.71 ± 2.75			
	Global Motricity	Yes	15	28.53 ± 4.71	0.945	0.00	0.131
		No	7	27.86 ± 5.42			
	Fine motor skills	Yes	15	19.33 ± 4.48	0.891	0.00	0.019
		No	7	19.43 ± 5.91			
36 months	Posture Skills	Yes	55	12.05 ± 2.52	0.356	0.01	0.282
		No	23	11.30 ± 2.78			
	Locomotive Skills	Yes	55	9.13 ± 1.71	0.410	0.00	0.217
		No	23	8.83 ± 0.93			
	Object Handling Skills	Yes	55	7.69 ± 1.67	0.085	0.03	0.485
		No	23	7.00 ± 1.12			
	Fine Manipulation Skills	Yes	55	10.56 ± 2.83	0.601	0.00	0.106
		No	23	10.91 ± 3.71			
	Skills VM integration	Yes	55	**11.56 ± 2.04**	**0.001**	**0.13**	**0.817**
		No	23	9.96 ± 1.87			
	Global Motricity	Yes	55	28.56 ± 3.95	0.373	0.01	0.246
		No	23	27.57 ± 4.08			
	Fine motor skills	Yes	55	**23.44 ± 4.26**	**0.006**	**0.09**	**0.673**
		No	23	20.57 ± 4.26			
48 months	Posture Skills	Yes	63	10.21 ± 2.26	0.606	0.00	0.117
		No	30	10.47 ± 2.17			
	Locomotive Skills	Yes	63	7.84 ± 1.50	0.292	0.01	0.256
		No	30	7.47 ± 1.38			
	Object Handling Skills	Yes	63	6.78 ± 0.94	0.672	0.00	0.022
		No	30	6.80 ± 0.80			
	Fine Manipulation Skills	Yes	63	10.24 ± 2.64	0.136	0.02	0.281
		No	30	11.00 ± 2.76			
	Skills VM integration	Yes	63	11.41 ± 1.88	0.685	0.00	0.095
		No	30	11.23 ± 1.88			
	Global Motricity	Yes	63	25.21 ± 4.10	0.954	0.00	0.064
		No	30	24.97 ± 3.27			
	Fine motor skills	Yes	63	21.51 ± 4.30	0.442	0.00	0.096
		No	30	21.87 ± 3.09			

Significant values are in bold. N—sample number; M—Mean; SD—Standard Deviation; Min—Minimum; Max—Maximum; ε^2^: 0–0.2: small; 0.21–0.6: medium; >0.61: large.

**Table 3 healthcare-13-03191-t003:** Differences between the variable Minutes per day of Technology use in PDMS-2 for each age group.

Access to Technologies (min/Day)	PDMS-2	Technologies (min/Day)	N	M ± SD	*p*	ε^2^	Effect Size
24 months	Posture Skills	<60	17	11.35 ± 2.12	0.880	0.02	0.074
		>60	5	11.20 ± 1.92			
	Locomotive Skills	<60	17	7.29 ± 1.82	0.120	0.01	0.962
		>60	5	9.40 ± 2.51			
	Object Handling Skills	<60	17	8.12 ± 1.79	0.401	0.01	0.606
		>60	5	9.00 ± 1.00			
	Fine Manipulation Skills	<60	17	9.71 ± 3.67	0.249	0.00	0.601
		>60	5	11.60 ± 2.51			
	Skills VM integration	<60	17	7.82 ± 2.55	0.189	0.01	0.633
		>60	5	9.60 ± 3.05			
	Global Motricity	<60	17	27.88 ± 5.23	0.446	0.01	0.446
		>60	5	29.80 ± 3.11			
	Fine motor skills	<60	17	19.24 ± 5.01	0.940	0.07	0.115
		>60	5	19.80 ± 4.65			
36 months	Posture Skills	<60	46	11.20 ± 1.92	0.328	0.01	0.200
		>60	32	11.65 ± 2.53			
	Locomotive Skills	<60	46	8.96 ± 2.53	0.154	0.02	0.103
		>60	32	9.16 ± 1.05			
	Object Handling Skills	<60	46	7.67 ± 1.72	0.313	0.01	0.298
		>60	32	7.22 ± 1.26			
	Fine Manipulation Skills	<60	46	10.67 ± 3.61	0.955	0.00	0.003
		>60	32	10.66 ± 2.20			
	Skills VM integration	<60	46	10.91 ± 2.16	0.269	0.15	0.204
		>60	32	11.34 ± 2.04			
	Global Motricity	<60	46	28.13 ± 3.85	0.691	0.00	0.083
		>60	32	28.47 ± 4.24			
	Fine motor skills	<60	46	22.13 ± 4.28	0.239	0.01	0.251
		>60	32	23.25 ± 4.62			
48 months	Posture Skills	<60	42	9.93 ± 2.46	0.208	0.01	0.332
		>60	51	10.59 ± 1.36			
	Locomotive Skills	<60	42	7.52 ± 1.36	0.238	0.01	0.247
		>60	51	7.88 ± 1.54			
	Object Handling Skills	<60	42	6.74 ± 1.03	0.230	0.01	0.088
		>60	51	6.82 ± 0.76			
	Fine Manipulation Skills	<60	42	10.43 ± 2.95	0.864	0.00	0.036
		>60	51	10.53 ± 2.48			
	Skills VM integration	<60	42	11.31 ± 1.91	0.736	0.00	0.042
		>60	51	11.39 ± 1.86			
	Global Motricity	<60	42	24.52 ± 4.04	0.152	0.02	0.289
		>60	51	25.63 ± 3.62			
	Fine motor skills	<60	42	21.14 ± 3.40	0.223	0.01	0.226
		>60	51	22.02 ± 4.32			

N—sample number; M—Mean; SD—Standard Deviation; Min—Minimum; Max—Maximum; ε^2^: 0–0.2: small; 0.21–0.6: medium; >0.61: large.

## Data Availability

The data presented in this study are available on request from the corresponding author due to data protection restrictions for minors.

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
