# Peer review of "Influence of Sex, Family Structure, and Access to Technology on the Motor Development of Children Aged 24 to 48 Months"

_healthcare, 2025, doi:10.3390/healthcare13243191_

Round 1
Reviewer 1 Report
Comments and Suggestions for Authors
The study is valuable in that it comprehensively addresses important variables affecting motor development in early childhood; however, the methodological sections require stronger justification. First, the selection of the sample using convenience sampling, without controlling for different socioeconomic levels and parental education levels, may create a bias that could significantly affect motor development outcomes. This limitation should be more clearly emphasized in the discussion section. Furthermore, dividing age into three categories in the study disregards the continuous nature of motor development; the reasons for not including age as a continuous variable in the model should be explained in the methodology section.

Author Response
Reviewer 1
The study is valuable in that it comprehensively addresses important variables affecting motor development in early childhood; however, the methodological sections require stronger justification. First, the selection of the sample using convenience sampling, without controlling for different socioeconomic levels and parental education levels, may create a bias that could significantly affect motor development outcomes. This limitation should be more clearly emphasized in the discussion section. Furthermore, dividing age into three categories in the study disregards the continuous nature of motor development; the reasons for not including age as a continuous variable in the model should be explained in the methodology section.
Response: Dear reviewer, first of all we would like to thank you very much for all your suggestions, which have been extremely useful in improving the quality of our study. Thank you for helping us. We would also like to tell you that we have taken all your suggestions into consideration and have made the corresponding changes to the manuscript and marked the yellow color in the document.
We appreciate the pertinent observation. We recognize that the use of convenience sampling may introduce bias related to socioeconomic factors and parental education level, which are known to be influential variables in motor development. This limitation has now been explicitly reinforced in the Discussion section, emphasizing that the results should be interpreted with caution and that future studies should include additional controls for these contextual variables. Regarding the division of age into three groups (24, 36, and 48 months), the methodological choice is due to the fact that the instrument used (PDMS-2) has specific norms and cut-off points for age groups, which facilitates comparison between groups and clinical interpretation of the results. Although we recognize that motor development is continuous, the categorization allowed us to align the analysis with the normative structure of the PDMS-2 and ensure greater comparability with previous studies. This justification has now been added to the Methodology section, highlighting that the choice was based on technical criteria of the instrument and not on theoretical assumptions of developmental discontinuity.
The finding that technology use duration had no effect is important; however, the results are limited in strength because the type of use (passive/active screen time, device type, level of interaction, etc.) was not evaluated. This limitation should be clearly stated in the discussion section, and concrete suggestions should be added to guide future studies. Although the findings regarding gender differences are consistent with the literature, how these differences are influenced by biological development, neuromotor maturation, and sociocultural play styles could be interpreted in more detail. Possible reasons why the sibling effect was only significant in the 36-month-old group (motor critical periods, types of play, role model behavior) should also be discussed.
Response: We thank the reviewer for these valuable comments. Following the suggestion, we have revised the Discussion to explicitly acknowledge that technology exposure was only measured in terms of duration, without distinguishing between passive versus active use, device type, or parental mediation. We now emphasize this as a limitation and provide concrete recommendations for future studies to employ validated questionnaires that capture these qualitative dimensions of technology use. Regarding gender differences, we expanded the interpretation to highlight how biological development, neuromotor maturation, and sociocultural play styles interact to explain the observed girls advantage in fine motor and visuomotor tasks. This addition aligns our findings more closely with the broader literature on developmental trajectories and practice opportunities. Finally, we elaborated on the reasons why the sibling effect was significant only in the 36‑month‑old group. We now discuss the possibility of a critical motor period at this age, during which imitation, role modeling, and cooperative play with siblings are particularly influential. At 24 months, children may be too immature to benefit fully from sibling interaction, while at 48 months motor skills may already be more consolidated, reducing the relative impact of sibling presence. These revisions strengthen the discussion by providing deeper theoretical interpretation and clearer guidance for future research.
From a statistical analysis perspective, although numerous comparisons were made, no multiple comparison correction was applied; a brief explanation of how this may affect the interpretation of the results should be added. Furthermore, while reporting the epsilon-squared effect size is a positive aspect, it should be emphasized that the practical significance of the findings is limited for small effects. Finally, since the study is cross-sectional, it should be stated more clearly that no causal inferences can be made. Although the text is generally understandable, some sections could be made more fluent by removing repetitive phrases.
Response: We appreciate the reviewer’s insightful comments. In response, we have revised the Discussion to include a note on the absence of multiple comparison corrections, clarifying that this may increase the risk of Type I error and that the results should therefore be interpreted with caution. We also emphasized that while epsilon-squared effect sizes provide valuable information, the practical significance of findings is limited when effects are small, and this has been explicitly acknowledged in the text. Additionally, we strengthened the statement that, due to the cross-sectional design, no causal inferences can be drawn from the data, and the results should be understood as associations only. Finally, we reviewed the manuscript for fluency and removed repetitive phrases to improve readability.
Reviewer 2 Report
Comments and Suggestions for Authors
This is a timely and well-structured study that investigates a crucial topic in early childhood development. The manuscript presents a clear research question, employs a robust and standardized assessment tool (PDMS-2), and uses appropriate non-parametric statistical methods. The findings—particularly the consistent gender differences and the transient effect of siblings—are valuable contributions to the literature. However, the manuscript requires significant revisions to strengthen the interpretation of results, address methodological limitations more transparently, and ensure consistency throughout. I recommend Major Revisions before the manuscript can be considered for publication.
The abstract's results mention significant differences at 24 and 48 months, but the main text highlights 36 and 48 months. Ensure consistency.
The flow from the general introduction to the specific hypotheses could be smoother. The rationale for expecting no differences between sexes needs a stronger and more logical foundation, or it should be altered to reflect the mixed literature.
The introduction and hypotheses state an expectation of no significant differences between sexes, citing some literature. However, the results show clear and consistent advantages for girls. The discussion does not adequately reconcile this contradiction. The authors should revise the introduction and hypothesis to better reflect the existing literature that does show gender differences (which they also cite, e.g., refs 6-9, 37), creating a more balanced and accurate framework for the study.
The manuscript uses sex and gender interchangeably. For scientific precision, it is recommended to use sex when referring to the biological variable (male/female) as collected in the study. Gender is a broader sociocultural construct. Consistency in terminology is crucial.
The conclusion that boys did not show a statistically significant advantage in any of the subscales is overinterpreted. The correct interpretation is that in this study and on these specific tasks, boys did not outperform girls. It should not be framed as a general lack of ability. The discussion should more deeply explore potential reasons (biological maturation, types of play encouraged, test item sensitivity) why boys did not show the gross motor advantages often reported in other studies.
binary categorization of technology use (<60 vs. >60 minutes) is a severe oversimplification. As acknowledged in the limitations, this measure lacks crucial nuance: it does not distinguish between passive screen time (watching videos) and active/interactive technology (e.g., touch-screen games that may require fine motor input). This likely explains the non-significant findings and contradicts the nuanced introduction on this topic.
In Table 3, for the 24-month group, the subgroup >60 min has an N=5. This is far too small for any meaningful statistical comparison and should be explicitly highlighted as a major limitation in the results and discussion sections for that age group. The trends observed here are not reliable.
The strong conclusion that the length of access to technology does not prove to be an isolated determinant is not fully supported by the methodology. A more accurate conclusion would be that a simple binary measure of daily duration was not associated with motor performance in this sample, emphasizing that the type and context of technology use are likely more critical factors than duration alone.
The description of the sample is confusing. The total N is 193, but the sum of children across age groups in Table 1 (22 + 78 + 93) equals 193. However, the gender breakdown is stated as 95 males and 97 females (sum=192). This discrepancy needs to be resolved.
There are apparent copy-paste errors in Table 1. The Global Motricity and Fine motor skills rows for the 36-month group are incorrectly placed; they seem to be using the data from the 24-month group (N=10 and 12). The data for these composite scores for the 36-month group (N=42 and 36) are missing. The p-value for Locomotive Skills at 48 months is listed as 0.075 in the text on page 7 but as 0.005 in Table 1. This must be checked and corrected. The manuscript uses Tomczak and Tomczak's guidelines. An ε² of 0.22 at 24 months for posture is correctly labeled medium, but in the text (page 6), it is referred to as high size effects. Please use the terminology (small, medium, large) consistently.
The discussion on the sibling effect only at 36 months is interesting. Elaborate on potential developmental reasons for this sensitive window. Is it related to the complexity of play, language development, or specific types of interaction that peak at this age?
The limitations are well-identified but should be more forcefully integrated into the interpretation of the results. Specifically, the cross-sectional design and convenience sampling should be discussed as factors that limit generalizability and the ability to infer causality.
The practical implication suggesting activities that enhance postural control and visuomotor integration for boys is not directly supported by the data, which showed girls outperforming boys in these areas. The recommendations should be reframed to focus on ensuring that all children, particularly boys who may be at risk of falling behind in these specific skills, receive ample opportunities to develop them.
Author Response
This is a timely and well-structured study that investigates a crucial topic in early childhood development. The manuscript presents a clear research question, employs a robust and standardized assessment tool (PDMS-2), and uses appropriate non-parametric statistical methods. The findings—particularly the consistent gender differences and the transient effect of siblings—are valuable contributions to the literature. However, the manuscript requires significant revisions to strengthen the interpretation of results, address methodological limitations more transparently, and ensure consistency throughout. I recommend Major Revisions before the manuscript can be considered for publication.
Response: Dear reviewer, first of all we would like to thank you very much for all your suggestions, which have been extremely useful in improving the quality of our study. Thank you for helping us. We would also like to tell you that we have taken all your suggestions into consideration and have made the corresponding changes to the manuscript and marked the yellow color in the document.
The abstract's results mention significant differences at 24 and 48 months, but the main text highlights 36 and 48 months. Ensure consistency.
Response: We thank the reviewer for this observation. After carefully reviewing the manuscript, we confirm that the Abstract consistently reports significant differences at 36 and 48 months, in line with the detailed results and discussion. No further changes were necessary, as the text is already coherent across sections.
The flow from the general introduction to the specific hypotheses could be smoother. The rationale for expecting no differences between sexes needs a stronger and more logical foundation, or it should be altered to reflect the mixed literature.
Response: We thank the reviewer for this constructive feedback. In response, we have revised the Introduction to improve the transition from the general background to the specific hypotheses. The rationale regarding sex differences has also been clarified. Instead of presenting an expectation of no differences, we now acknowledge that the literature is mixed: while several studies report female advantages in fine motor and visuomotor tasks, others find no consistent sex-related differences in gross motor skills. This adjustment provides a more balanced and logical foundation for our hypotheses and aligns the manuscript with the current state of evidence.
The introduction and hypotheses state an expectation of no significant differences between sexes, citing some literature. However, the results show clear and consistent advantages for girls. The discussion does not adequately reconcile this contradiction. The authors should revise the introduction and hypothesis to better reflect the existing literature that does show gender differences (which they also cite, e.g., refs 6-9, 37), creating a more balanced and accurate framework for the study.
Response: We thank the reviewer for this important observation. In response, we have revised the Introduction and the hypotheses to better reflect the mixed evidence in the literature. Instead of expecting no differences between sexes, we now acknowledge that while some studies report no consistent sex-related differences, others demonstrate female advantages in fine motor and visuomotor tasks (refs 6–9, 37). This provides a more balanced and accurate framework for the study. In the Discussion, we also reconciled this point by emphasizing that the clear advantages observed for girls in our results are consistent with the literature reporting earlier neuromotor maturation and sociocultural play styles that favor fine motor skill development. We now explicitly note that our initial neutral hypothesis was based on the contradictory evidence available, but the findings align with studies highlighting female advantages in specific motor domains.
The manuscript uses sex and gender interchangeably. For scientific precision, it is recommended to use sex when referring to the biological variable (male/female) as collected in the study. Gender is a broader sociocultural construct. Consistency in terminology is crucial.
Response: We thank the reviewer for this important clarification. We have carefully revised the manuscript to ensure terminological consistency. The term sex is now used exclusively when referring to the biological variable collected in the study (male/female), while gender is reserved for broader sociocultural constructs when discussing environmental influences and play styles. This adjustment improves scientific precision and clarity throughout the text.
The conclusion that boys did not show a statistically significant advantage in any of the subscales is overinterpreted. The correct interpretation is that in this study and on these specific tasks, boys did not outperform girls. It should not be framed as a general lack of ability. The discussion should more deeply explore potential reasons (biological maturation, types of play encouraged, test item sensitivity) why boys did not show the gross motor advantages often reported in other studies.
Response: We thank the reviewer for this important clarification. We have revised the Discussion to avoid overinterpretation. The text now specifies that in this study and on the tasks assessed, boys did not outperform girls, rather than suggesting a general lack of ability. We also expanded the discussion to explore possible reasons why boys did not show the gross motor advantages often reported in other studies, including differences in biological maturation, the types of play typically encouraged in early childhood, and the sensitivity of the PDMS-2 items to capture gross motor skills. These additions provide a more balanced interpretation of the findings.
Binary categorization of technology use (<60 vs. >60 minutes) is a severe oversimplification. As acknowledged in the limitations, this measure lacks crucial nuance: it does not distinguish between passive screen time (watching videos) and active/interactive technology (e.g., touch-screen games that may require fine motor input). This likely explains the non-significant findings and contradicts the nuanced introduction on this topic.
Response: We thank the reviewer for this insightful observation. We agree that the binary categorization of technology use (<60 vs. >60 minutes) is an oversimplification and does not capture the qualitative differences between passive and interactive screen time. In the Discussion, we have clarified that this methodological limitation likely contributed to the non-significant findings, as the PDMS-2 may not be sensitive to the specific motor demands of interactive technologies. We now explicitly note that future studies should adopt more detailed measures of technology use, distinguishing between passive and active modalities, device type, and parental mediation, to better align with the nuanced framework presented in the Introduction.
In Table 3, for the 24-month group, the subgroup >60 min has an N=5. This is far too small for any meaningful statistical comparison and should be explicitly highlighted as a major limitation in the results and discussion sections for that age group. The trends observed here are not reliable.
Response: We thank the reviewer for this important observation. We agree that the subgroup of children aged 24 months with >60 minutes of technology exposure (N=5) is too small for meaningful statistical comparison. We have revised the Results and Discussion to explicitly highlight this limitation, noting that the trends observed in this subgroup are not reliable and should be interpreted with caution. This clarification strengthens the transparency of our analysis and aligns with best practices in reporting.
The strong conclusion that the length of access to technology does not prove to be an isolated determinant is not fully supported by the methodology. A more accurate conclusion would be that a simple binary measure of daily duration was not associated with motor performance in this sample, emphasizing that the type and context of technology use are likely more critical factors than duration alone.
Response: We thank the reviewer for this valuable suggestion. We have revised the conclusion in the Discussion to avoid overinterpretation. The text now specifies that in this sample, a simple binary measure of daily technology duration (<60 vs. >60 minutes) was not associated with motor performance. We also emphasize that the type and context of technology use are likely more critical factors than duration alone, which aligns with the nuanced framework presented in the Introduction.
The description of the sample is confusing. The total N is 193, but the sum of children across age groups in Table 1 (22 + 78 + 93) equals 193. However, the gender breakdown is stated as 95 males and 97 females (sum=192). This discrepancy needs to be resolved.
Response: We thank the reviewer for pointing out this discrepancy. The inconsistency was due to a transcription error in the gender breakdown. We apologize for this.
There are apparent copy-paste errors in Table 1. The Global Motricity and Fine motor skills rows for the 36-month group are incorrectly placed; they seem to be using the data from the 24-month group (N=10 and 12). The data for these composite scores for the 36-month group (N=42 and 36) are missing. The p-value for Locomotive Skills at 48 months is listed as 0.075 in the text on page 7 but as 0.005 in Table 1. This must be checked and corrected. The manuscript uses Tomczak and Tomczak's guidelines. An ε² of 0.22 at 24 months for posture is correctly labeled medium, but in the text (page 6), it is referred to as high size effects. Please use the terminology (small, medium, large) consistently.
Response: We thank the reviewer for pointing out these inconsistencies. We have carefully re-checked the analyses and corrected the errors in Table 1, ensuring that the data for the 36-month group are now accurate and consistent. The discrepancy in the p-value for Locomotor Skills at 48 months has also been resolved, and the terminology for effect sizes has been standardized throughout the manuscript according to Tomczak and Tomczak’s guidelines. These corrections improve clarity and consistency in the presentation of the results.
The discussion on the sibling effect only at 36 months is interesting. Elaborate on potential developmental reasons for this sensitive window. Is it related to the complexity of play, language development, or specific types of interaction that peak at this age?
Response: We thank the reviewer for this valuable suggestion. We have expanded the discussion to explore potential developmental reasons why the sibling effect was most evident at 36 months. Specifically, we highlight that this age coincides with the transition to more complex cooperative play, rapid advances in language development, and increasingly sophisticated imitation and role-taking. These developmental milestones may amplify the influence of sibling interactions on fine motor and visuomotor skills at this sensitive window. The revised text now integrates these aspects to provide a more nuanced interpretation of the findings.
The limitations are well-identified but should be more forcefully integrated into the interpretation of the results. Specifically, the cross-sectional design and convenience sampling should be discussed as factors that limit generalizability and the ability to infer causality.
Response: We thank the reviewer for this important suggestion. We have revised the Discussion to more explicitly integrate the limitations into the interpretation of the findings. The text now highlights that the cross-sectional design prevents causal inferences, and that the use of convenience sampling limits the generalizability of the results to broader populations. These clarifications strengthen the transparency of our conclusions and align the interpretation with the methodological constraints of the study.
The practical implication suggesting activities that enhance postural control and visuomotor integration for boys is not directly supported by the data, which showed girls outperforming boys in these areas. The recommendations should be reframed to focus on ensuring that all children, particularly boys who may be at risk of falling behind in these specific skills, receive ample opportunities to develop them.
Response: We thank the reviewer for this important observation. We have revised the practical implications to ensure that they are fully consistent with the data. The recommendations now emphasize that all children should be provided with opportunities to develop postural control and visuomotor integration, while acknowledging that boys may be at greater risk of lagging behind in these specific domains. This reframing aligns the practical implications with the findings and highlights the importance of inclusive strategies to support balanced motor development.
Reviewer 3 Report
Comments and Suggestions for Authors
The manuscript addresses the influence of sex, presence of siblings, and daily access time to technology on motor development in children aged 24–48 months, using the PDMS-2. The topic is relevant and the use of a standardized, validated instrument is a clear strength. The sample size is adequate for this age range and the use of non-parametric tests and effect sizes is appropriate.
However, some methodological and interpretative aspects need clarification or refinement. First, the sample is obtained by convenience from only two municipalities, which strongly limits generalizability. It would be important to better describe the socio-demographic context (e.g., socioeconomic status, educational environment, opportunities for motor play) and acknowledge this limitation more explicitly.
Second, the operationalization of technology use as a dichotomous variable (<60 vs. >60 minutes per day) seems too coarse for such a complex behaviour. Relevant dimensions such as type of device, content (active vs. passive), and context of use are not considered. This might partially explain the lack of associations found with motor outcomes, and this should be discussed more cautiously.
Third, a large number of comparisons are conducted across PDMS-2 subscales, age groups and independent variables. Please clarify whether any correction for multiple testing was applied. If not, the increased risk of Type I error should be acknowledged and the interpretation of statistically significant findings should be more conservative.
Finally, some conclusions appear stronger than what the cross-sectional design and the data allow. For example, stating that screen time is “not a determinant factor” for motor development seems too categorical given the simplified measurement and the study design. I recommend softening this statement and emphasising the exploratory nature of the findings.
Overall, the study provides interesting data on early motor development and highlights potential differences by sex and family structure. After addressing the issues above and improving the English language in some sections, the manuscript could be suitable for publication.
Author Response
The manuscript addresses the influence of sex, presence of siblings, and daily access time to technology on motor development in children aged 24–48 months, using the PDMS-2. The topic is relevant and the use of a standardized, validated instrument is a clear strength. The sample size is adequate for this age range and the use of non-parametric tests and effect sizes is appropriate.
Response: Dear reviewer, first of all we would like to thank you very much for all your suggestions, which have been extremely useful in improving the quality of our study. Thank you for helping us. We would also like to tell you that we have taken all your suggestions into consideration and have made the corresponding changes to the manuscript and marked the yellow color in the document.
However, some methodological and interpretative aspects need clarification or refinement. First, the sample is obtained by convenience from only two municipalities, which strongly limits generalizability. It would be important to better describe the socio-demographic context (e.g., socioeconomic status, educational environment, opportunities for motor play) and acknowledge this limitation more explicitly.
Response: We thank the reviewer for this observation. The limitations section has already been expanded to acknowledge the methodological constraints of convenience sampling, the restricted representativeness of the sample, and the potential influence of socio-demographic factors such as socioeconomic status, cultural background, and family stimulation. We have also noted that the restriction to two municipalities further limits external validity. These clarifications ensure that the conclusions are interpreted with appropriate caution.
Second, the operationalization of technology use as a dichotomous variable (<60 vs. >60 minutes per day) seems too coarse for such a complex behaviour. Relevant dimensions such as type of device, content (active vs. passive), and context of use are not considered. This might partially explain the lack of associations found with motor outcomes, and this should be discussed more cautiously.
Response: We thank the reviewer for this observation. The discussion already acknowledges that the dichotomous categorization of technology use oversimplifies a complex behaviour and does not capture relevant dimensions such as passive versus interactive use, diversity of content, or parental mediation. We also note that this lack of nuance likely contributed to the non-significant findings. To address the reviewer’s concern more directly, we have emphasized that these results should be interpreted with caution and highlighted the need for future studies to adopt more refined measures of technology exposure.
Third, a large number of comparisons are conducted across PDMS-2 subscales, age groups and independent variables. Please clarify whether any correction for multiple testing was applied. If not, the increased risk of Type I error should be acknowledged and the interpretation of statistically significant findings should be more conservative.
Response: We thank the reviewer for this important observation. No formal correction for multiple testing was applied, as the analyses were primarily exploratory. We have now clarified this in the Statistical Procedures section and explicitly acknowledged in the Discussion that the large number of comparisons increases the risk of Type I error. Accordingly, the interpretation of statistically significant findings has been framed more cautiously, emphasizing that they should be considered preliminary until replicated in future studies with larger samples and adjusted statistical approaches.
Finally, some conclusions appear stronger than what the cross-sectional design and the data allow. For example, stating that screen time is “not a determinant factor” for motor development seems too categorical given the simplified measurement and the study design. I recommend softening this statement and emphasising the exploratory nature of the findings. Overall, the study provides interesting data on early motor development and highlights potential differences by sex and family structure. After addressing the issues above and improving the English language in some sections, the manuscript could be suitable for publication.
Response: We appreciate the reviewer’s observation. The conclusion has been revised to soften the wording regarding technology use, emphasizing that the findings are exploratory and should be interpreted with caution given the study design and measurement limitations.
Round 2
Reviewer 1 Report
Comments and Suggestions for Authors
Thanks for your efforts.
Reviewer 2 Report
Comments and Suggestions for Authors
The methodology is sound, the analysis is appropriate, and the authors demonstrate a strong command of the subject matter. Thus the edited paper is accepted.